

# Identifying stably expressed genes from multiple RNA-Seq data sets

Bin Zhuo[1], Sarah Emerson[1], Jeff H. Chang[2,3] and Yanming Di[1,3]

[1] Department of Statistics, Oregon State University, Corvallis, OR, United States
[2] Department of Botany and Plant Pathology, Oregon State University, Corvallis, OR, United States
[3] Molecular and Cellular Biology Graduate Program, Oregon State University, Corvallis, OR, United States of America

## ABSTRACT

We examined RNA-Seq data on 211 biological samples from 24 different Arabidopsis experiments carried out by different labs. We grouped the samples according to tissue types, and in each of the groups, we identified genes that are stably expressed across biological samples, treatment conditions, and experiments. We fit a Poisson log-linear mixed-effect model to the read counts for each gene and decomposed the total variance into between-sample, between-treatment and between-experiment variance components. Identifying stably expressed genes is useful for count normalization and differential expression analysis. The variance component analysis that we explore here is a first step towards understanding the sources and nature of the RNA-Seq count variation. When using a numerical measure to identify stably expressed genes, the outcome depends on multiple factors: the background sample set and the reference gene set used for count normalization, the technology used for measuring gene expression, and the specific numerical stability measure used. Since differential expression (DE) is measured by relative frequencies, we argue that DE is a relative concept. We advocate using an explicit reference gene set for count normalization to improve interpretability of DE results, and recommend using a common reference gene set when analyzing multiple RNA-Seq experiments to avoid potential inconsistent conclusions.

## INTRODUCTION

RNA sequencing (RNA-Seq) has become the technology of choice for transcriptome profiling over the last few years. The exponential growth in RNA-Seq studies have produced a large amount of *Arabidopsis thaliana* (Arabidopsis) data under a variety of experimental/environmental conditions. It is only natural to begin exploring how the large amount of existing data sets can help the analysis of future data. In this paper, we discuss identifying stably expressed genes from multiple existing RNA-Seq data sets based on a numerical measure of stability. We envision that such identified stably expressed genes could be used as a reference set or prior information for count normalization and differential expression (DE) analysis of future RNA-Seq data sets obtained from similar or comparable experiments. We also fit a random-effect model to the read counts for each gene and decompose the total variance into between-sample, between-treatment and

Corresponding author
Bin Zhuo, zhuob@oregonstate.edu

between-experiment variance components. The variance component analysis is a first step towards understanding the sources and nature of the RNA-Seq count variation. To illustrate our methods, we examined RNA-Seq data on 211 Arabidopsis samples from 24 different experiments carried out by different labs and identified genes that were stably expressed across biological samples, experimental or environmental conditions, and experiments (labs).

A reference set of stably-expressed genes will be useful for count normalization. A key task of RNA-Seq analysis is to detect DE genes under various experimental or environmental conditions. Count normalization is needed to adjust for differences in sequencing depths or library sizes (total numbers of mapped reads for each biological sample) due to chance variation in sample preparation. In DE analysis, gene expression levels are often estimated from relative read frequencies. For this reason, normalization is also needed to account for the fact that non-differentially expressing genes may exhibit an apparent reduction or increase in relative read frequencies due to the respective increased or decreased relative read frequencies of truly differentially expressing genes. Many existing normalization methods, such as the trimmed mean of $M$-values normalization method (TMM) (*Robinson & Oshlack, 2010*) and Anders and Huber's normalization (*Anders & Huber, 2010*), assume that the majority of the genes within an experiment are not DE, and examine the sample distribution of the fold changes between samples. If the experiment condition can affect expression levels of more than half of the genes, many of the existing normalization methods may be unreliable (*Lovén et al., 2012*; *Wu et al., 2013*). This difficulty could be alleviated if one could identify a set of stably expressed genes whose expression levels are known or expected to not vary much under different experimental conditions. Our idea is to identify such a reference set based on a large number of existing data sets.

Our basic intuition is that a numerical quantification of expression stability—which typically measures certain aspects of RNA-Seq count variation—can be more reliably estimated by using more data sets. There is, however, a caveat to this idea: as pointed out by *Fernandes et al. (2008)* and *Hruz et al. (2011)*, universally stably expressed genes may not exist. *Hruz et al.* showed that a subset of stably expressed genes from a specific biological context may have more variability than other genes if examined across a broader range of samples and conditions. Many studies have shown that stably expressed genes are subject to change from one experiment to another due to different experimental protocols, different tissue types, or other varying conditions (*Hong et al., 2010*; *Reid et al., 2006*). The top 100 stably expressed genes in the Arabidopsis developmental series of *Czechowski et al. (2005)* shared only 3 genes with the top 50 stably expressed genes identified from Arabidopsis seed samples by *Dekkers et al. (2012)*. In this study, we try to balance generality and specificity by identifying different reference gene sets for different tissue types of Arabidopsis.

We can also consider that when a normalization method is applied to a single data set, it effectively specifies an implicit reference set of stably expressed genes (those genes that have the least variation after normalization). From this perspective, we can view commonly used normalization techniques as using an internally identified reference set of genes. In contrast, what we are proposing is that one could alternatively identify a reference set externally by looking at past data sets. The internally and externally identified reference

gene sets will provide different contexts for the DE analysis: in other words, one can choose to answer different scientific questions by using different reference sets. In any case, we advocate making the reference set explicit during a DE analysis and using a common reference set when analyzing multiple datasets.

We want to clarify that having stable gene expression is not equivalent to maintaining a stable biological function. Often times, we may not understand the biological functions of genes with numerically stable expression measures. From an operational point of view, however, numerical stability is more tractable. In the pre-genomic era, the so-called "*house-keeping genes*" were often considered to be candidate reference genes for normalization (*Andersen, Jensen & Orntoft, 2004*; *Bustin, 2002*). House-keeping genes are typically constitutive genes that maintain basic cellular function, and therefore are expected to express at relatively constant levels in non-pathological situations. However, many studies have shown that house-keeping genes are not necessarily stably expressed according to numerical measures (a review can be found in *Huggett et al. (2005)* and reference therein). For example, in the microarray analysis of Arabidopsis, *Czechowski et al. (2005)* showed that traditional house-keeping genes such as ACT2, TUB6, EF-1$\alpha$ are not stably expressed, and thus not good reference genes for normalization. Spike-in genes have also been considered as reference genes for normalization, but *Risso et al. (2014)* showed that spike-in genes are not necessarily stably expressed according numerical measures either.

In this paper, we identify stably expressed genes from RNA-Seq data sets based on a numerical measure—the sum of three variance components estimated from a mixed-effect model. For microarray data, there have been many efforts to numerically find stably expressed genes by quantifying the variation of measured expression levels across a large number of microarray data sets. For example, *Andersen, Jensen & Orntoft (2004)* used a linear mixed model to estimate the between-group and within-group variances from expression profiles of microarray experiments, and then quantified expression stability by combining the two variance components using a Bayesian formulation. *Czechowski et al. (2005)* measured the expression stability of each gene using the coefficient of variation (CV). Genes with lower CVs are considered more stably expressed. By investigating 721 arrays under 323 conditions throughout development, *Czechowski et al. (2005)* suggested stably expressed (reference) genes under different experimental conditions for Arabidopsis. *Stamova et al. (2009)*, *Dekkers et al. (2012)*, *Gur-Dedeoglu et al. (2009)*, and *Frericks & Esser (2008)* screened a large number of microarray data sets to identify stably expressed genes in human blood, Arabidopsis seed, breast tumor tissues, and mice respectively. Validation experiments (*Czechowski et al., 2005*; *Dekkers et al., 2012*; *Huggett et al., 2005*; *Stamova et al., 2009*) showed that these genes are more stably expressed than traditional house-keeping genes.

Our vision is that identifying stably expressed genes is the first step towards integrative analysis of multiple RNA-Seq experiments. It will help to answer fundamental questions related to comparability, reproducibility and replicability of RNA-Seq experiments.
**Table 1  Summary statistics for the three groups of Arabidopsis samples.**

| Group | # Experiments | # Treatments | # Samples | # Genes |
|---|---|---|---|---|
| Seedling | 9 | 27 | 60 | 22,207 |
| Leaf | 5 | 28 | 60 | 20,967 |
| Multi-tissue | 10 | 39 | 91 | 23,611 |

# MATERIALS & METHODS

In 'RNA-Seq data collection and processing', we describe the steps for collecting and processing RNA-Seq data sets from Arabidopsis experiments. In 'Count normalization', we discuss count normalization methods and how to apply them to a subset of stably expressed genes. In 'Poisson log-linear mixed-effects regression model and the total variance measure of expression stability', we introduce the generalized linear mixed model (GLMM, *McCulloch & Neuhaus, 2001*) for estimating three variance components from RNA-Seq data: the *between-sample*, *between-treatment* and *between-experiment* variances. We define the *total variance* measure for expression stability as the sum of estimated variance components. In 'Other stability measures', we review the CV and *M*-value measures for gene expression stability.

## RNA-Seq data collection and processing
### Overview of the RNA-Seq data sets

We examined RNA-Seq data from 49 Arabidopsis experiments stored on the NCBI GEO repository (see more details below). After screening, we retained data from 211 biological samples in 24 experiments. To illustrate our methods for finding stably expressed genes, we divided the experiments into three groups: *the seedling group* contains 60 Arabidopsis seedling samples from 9 experiments; *the leaf group* contains 60 Arabidopsis leaf samples from 5 experiments; the *multi-tissue group* contains 91 samples from 10 experiments on multiple tissue types (shoot apical, root tip, primary root, inflorescences and siliques, hypocotyl, flower, carpels, aerial tissue, epidermis, seed). Table 1 summarizes the basic information about the three groups (see Table S1 for more details).

To find stably expressed genes in each group, we processed the raw sequencing data and summarized the results as count matrices of mapped RNA-Seq short reads (see details below). We removed genes with low mean numbers (less than 3) of mapped read counts for all experiments. Such genes tend to be more prone to sequencing noise, less interesting to biologists, and also cause convergence issues when fitting statistical models. Many other researchers (such as *Anders et al., 2013*) recommend removing such genes before analyzing RNA-Seq data. The number of remaining genes in each group is also summarized in Table 1. Figure 1 shows the numbers and overlap of the genes after this step.

### Details of the data processing steps
The Gene Expression Omnibus (GEO) repository at National Center for Biotechnology Information (NCBI, http://www.ncbi.nlm.nih.gov/) stores raw sequencing data from a large number of RNA-Seq experiments. For this study, we restrict our attention to Arabidopsis experiments satisfying the following conditions: 1. Ecotype = ''Columbia''

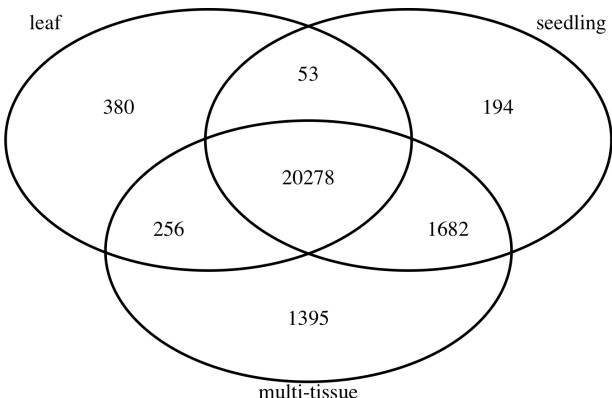

**Figure 1** The numbers and overlap of the genes in the three groups of Abrabidopsis samples after removing genes with low mean counts.

(we kept only the Columbia samples from experiments that compare Columbia samples to other ecotypes); 2. There are at least two treatments and 2 biological replicates for each treatment; 3. Library strategy = "RNA-Seq"; 4. Library source = "transcriptomic"; 5. Library selection = "cDNA"; 6. Library layout = "Single end"; 7. If there are repeated measurements over time, we choose samples from one time point. We screened all the Arabidopsis experiments available from the NCBI GEO repository up to May 31, 2015 and downloaded raw RNA-Seq data (Sequence Read Archive files) from 49 experiments.

We assembled our own in-house pipeline to process all the raw RNA-Seq data: align the raw RNA-Seq reads to the reference genome and summarize the read counts at the gene level. In the GEO repository, the mapped read counts are unavailable for some experiments and the available ones are from different processing pipelines. Our pipeline, implemented using the software R (*R Core Team, 2015*), is summarized as follows:

1. Convert the Sequence Read Archive (SRA) files to FASTQ files using the NCBI SRA Toolkit (*Leinonen, Sugawara & Shumway (2010)*, version 2.3.5-2).

2. Download the reference genome

   ```
   Arabidopsis_thaliana.TAIR10.22.dna.toplevel.fa
   ```

   from the *Ensembl plants FTP server* (http://plants.ensembl.org/info/data/ftp/index. html) and build index using `build()` function from Subread aligner (RSubread, version 1.16.2, *Liao, Smyth & Shi, 2013*) in the software R (*R Core Team, 2015*). The index allows fast retrieval of the sets of positions in the reference genome where the short reads are more likely to align.

3. Align short reads in FASTQ files to the Arabidopsis reference genome using the `align()` function from Rsubread.

4. Summarize the read counts at the gene level using the `featureCounts()` function from the Subread aligner and store the read counts as data matrix. The annotation file

   ```
   Arabidopsis_thaliana.TAIR10.22.gtf
   ```

is downloaded from Ensembl plants FTP server. To keep the pipeline simple, we did not count multi-mapping or multi-overlapping reads. One potential challenge when dealing with multi-mapping reads is that existing methods will assign reads to different gene features proportionally and probabilistically, it is unclear to us how to handle the additional uncertainty associated with such a process (see, e.g., *Anders, Pyl & Huber (2014)*). DE analysis of multiple mapped reads often requires special method.

Subread aligner is a recently developed sequence mapping tool that adopts a seed-and-vote paradigm to map the RNA-Seq short reads to the genome. It breaks each short read into a series of overlapping segments called subreads and uses the subreads to vote on the optimal genome location of the original read. The subreads are shorter and can be mapped to the genome much faster. Compared to other aligners such as Bowite 2 (*Langmead & Salzberg, 2012*) or BWA (*Li & Durbin, 2009*), Subread aligner is both faster and more accurate (*Liao, Smyth & Shi, 2013*). We compared results from the above pipeline to results from a pipeline described in *Anders et al. (2013)* over several RNA-Seq experiment data, and Rsubread was more than three times faster and successfully mapped more reads to the reference genome. For researchers familiar with R, it also has the advantage that it is completely implemented in R.

We divided the experiments into three groups as summarized in Table 1. As an additional data quality control measure, we keep an experiment only when it has mapping quality (number of successfully mapped reads divided by total number of reads) $\geq 50\%$ for all samples. Then within each group, we computed an initial set of normalization factors from all samples combined using the method described in 'Count normalization'. An experiment is retained only when the normalization factors of all samples in the experiment are between 0.50 and 1.50. If the initial estimated normalization factor is too different from 1 for a sample, it often indicates that the read counts distribution in the corresponding sample is markedly different from the distributions of the rest of the samples. Such samples demand additional attention before being incorporated in the studies that we intend to do.

## Count normalization

As explained in the introduction, count normalization is needed when analyzing RNA-Seq data to (1) adjust for differences in sequencing depths or library sizes; (2) to adjust for the apparent changes in relative read frequencies of non-DE genes that occur as a consequence of changes in relative read frequencies of truly DE genes.

For the second type of adjustment, we follow Anders and Huber's method (*Anders & Huber, 2010*) for estimating normalization factors. Let $y_{ij}$ denote the read count for $i$th gene of the $j$th sample ($m$ genes and $n$ samples in total). We first create a pseudo-reference sample where each gene's expression value is the geometric mean expression over all real samples for that gene,

$$y_{i,0} = (\prod_{j=1}^{n} y_{i,j})^{1/n}, i = 1, \ldots, m. \tag{1}$$

Next we calculate the median fold-change in relative frequency between each sample $j$ and the pseudo-reference sample,

$$R'_j = \text{median}\left(\frac{y_{1,j}/N_j}{y_{1,0}/N_0}, \dots, \frac{y_{m,j}/N_j}{y_{m,0}/N_0}\right), \tag{2}$$

where $N_j$ is the library size for sample $j$ (the sum of RNA-Seq counts mapped to all genes retained in each sample). Finally, the *normalization factor $R_j$* for sample $j$ is calculated as

$$R_j = \frac{R'_j}{(\prod_{j=1}^{n} R'_j)^{1/n}}. \tag{3}$$

Using the estimated normalization factors, the relative frequencies will be computed as $y_{ij}/N_j R_j$, which we will call the *normalized relative frequency* for gene $i$ in sample $j$. The assumption made here is that the median fold change between normalized relative frequencies in two samples should be 1. In other words, this normalization method assumes that the majority of genes are not DE. The NBPSeq package (*Di, Schafer & Di, 2013*) has an inbuilt function for this procedure and it will be used for count normalization in this paper. With the estimates from Eq. (3), we see that the median fold change in normalized relative frequencies between each sample and the pseudo-reference sample will be set to 1:

$$\text{median}\left(\frac{y_{1,j}/N_j R_j}{y_{1,0}/N_0 R_0}, \dots, \frac{y_{m,j}/N_j R_j}{y_{m,0}/N_0 R_0}\right) = 1, \tag{4}$$

where $R_0 = (\prod_{j=1}^{n} R'_j)^{-1/n}$.

We can apply Eq. (2) to a subset of reference genes to estimate normalization factors. In doing so, effectively, the median fold change in Eq. (4) among the reference genes will be set to 1 in each sample $j$. Other normalization methods may make different assumptions than Anders and Huber's, but some assumptions of a similar nature seem unavoidable. For example, the TMM method of *Robinson & Oshlack (2010)* is based on a similar principle: assuming the majority of the genes are not DE. The TMM method can be applied to a subset of genes selected based on an initial screening of mean expression level and fold changes. In TMM method, one can also specify certain quantile (instead of the median) of the fold changes to be 1.

In this paper, we will identify stably expressed genes from multiple data sets based on numerical measure and use them as reference for estimating normalization factors (from Eqs. (2) and (3)). However, to identify the stably expressed genes, we first need a set of initially estimated normalization factors. To tackle this circular dependence, we use a one-step iteration method to estimate the normalization factors:

1. First, we use all the genes to calculate the initial normalization factors;
2. Then, we fit a GLMM to each gene and estimate the total variance measure, incorporating the initial normalization factors as an offset term (see 'Poisson log-linear mixed-effects regression model and the total variance measure of expression stability');
3. Next, we select the top 1,000 stably expressed genes based on the total variance measure estimated from step 2 above, and use them as reference genes to recalculate the normalization factors.

In practice, this one-step method seems to be adequate and further iterations will only slightly change the set of 1,000 stably expressed genes. For example, for the multi-tissue group of experiments, if we were to run one more iteration of steps 2 and 3, there would be 946 overlapping genes between the top 1,000 genes from the first iteration and those from the second iteration.

## Poisson log-linear mixed-effects regression model and the total variance measure of expression stability

We fit a Poisson log-linear mixed-effects regression model to the RNA-Seq counts mapped to each gene and measure gene expression stability using a total variance measure. Let $Y_{ijkl}$ be the number of RNA-Seq reads mapped to gene $i$ in sample $j$ from treatment group $k$ in experiment $l$. We will fit regression models to each gene separately and suppress subscript $i$ from the model equations. For each gene, we fit a Poisson log-linear mixed-effects regression model

$$Y_{jkl} \sim \text{Poisson}(\mu_{jkl}), \tag{5}$$

$$\log(\mu_{jkl}) = \log(R_{jkl} N_{jkl}) + \xi + \alpha_l + \beta_{k(l)} + \epsilon_{jkl}, \tag{6}$$

which is a specific type of generalized linear mixed model (GLMM, *McCulloch & Neuhaus, (2001)*). In Eq. (6), $N_{jkl}$ and $R_{jkl}$ are the library size and normalization factor discussed in 'Count normalization'. We will call $R_{jkl} N_{jkl}$ the *normalized library size*. The parameter $\xi$ is a fixed-effect term for the baseline log mean of the *relative counts* (counts divided by the normalized library sizes). The values $\alpha$, $\beta$, and $\epsilon$ represent the experiment effect, the treatment effect (nested within each experiment), and the sample effect respectively. We view $\alpha$, $\beta$ and $\epsilon$ as random effects and assume that they are independent and follow normal distributions:

$$\alpha_l \sim N(0, \sigma^2_{\text{experiment}}), \beta_{k(l)} \sim N(0, \sigma^2_{\text{treatment}}), \epsilon_{jkl} \sim N(0, \sigma^2_{\text{sample}}), \tag{7}$$

where $\sigma^2_{\text{experiment}}$, $\sigma^2_{\text{treatment}}$ and $\sigma^2_{\text{sample}}$ are called *variance-components*—they quantify the overall variances of the corresponding random effect terms.

The sample effect $\epsilon$ represents the extra-Poisson variation in read counts among samples in the same treatment group and $\sigma^2_{\text{sample}}$ plays a similar role as the *over-dispersion* parameter in a negative binomial model (*Anders & Huber, 2010*; *Di et al., 2011*). The experiment effect, $\alpha$, accounts for all sources of variation at the experiment level, including differences in lab personnel and conditions, day light hours, age of the plants, temperature, sequencing platform, and other unidentified sources. The contributions from these different experiment-level sources are often difficult to separate statistically. We treat the experiment effect $\alpha$ as a random effect because we view the collected experiments as a random sample from the pool of all Arabidopsis RNA-Seq experiments. We also treat the treatment effect $\beta$ as a random effect. In a DE test, $\beta$ is usually considered as a fixed-effect term. Here for evaluation of expression stability, we are not interested in the specific levels of the individual $\beta$'s and focus more on the overall variation of $\beta$ under a range of treatment types.

We define the stability measure as the estimated *total variance*,

$$\hat{\sigma}^2 = \hat{\sigma}^2_{\text{sample}} + \hat{\sigma}^2_{\text{treatment}} + \hat{\sigma}^2_{\text{experiment}}. \tag{8}$$

The parameters $(\xi, \sigma^2_{\text{experiment}}, \sigma^2_{\text{treatment}}, \sigma^2_{\text{sample}})$ are estimated using the `glmer()` function of the R package `lme4` (*Bates, Mächler & Bolker (2015)*, version 1.1.7), which uses a Gaussian-Hermite quadrature to approximate the likelihood function. We rank all the genes according to their values of $\hat{\sigma}^2$ in increasing order (smallest first), and consider highly ranked (e.g., top 1,000) genes to be stably expressed.

Normal models (Eq. (7)) are commonly assumed for the random effects in the GLMM settings. The normality assumption is likely a simplification of reality, yet it is a good starting point and should be adequate for finding genes with low total variation—the stably expressed ones.

### Other stability measures

The assessment of gene expression stability depends on the specific stability measure used. *Czechowski et al. (2005)* and *Dekkers et al. (2012)* used the coefficient of variation (CV) measure, computed as *standard deviation divided by mean*, to find stably expressed genes from microarray data.

The $M$-value in geNorm (*Vandesompele et al., 2002*) is a well-cited measure. For a set of $m_0$ genes, the $M$-value measure works as follows: First, the *pairwise variation* between gene $i_1$ and gene $i_2$ is calculated as the standard deviation of the log fold changes between their expression levels across all the $n$ samples:

$$V_{i_1, i_2} = st.dev \left\{ \log\left(\frac{y_{1,i_1}}{y_{1,i_2}}\right), \ldots, \log\left(\frac{y_{n,i_1}}{y_{n,i_2}}\right) \right\}.$$

Next, the $M$-value for gene $i$ is defined as the average pairwise variation between gene $i$ and all other genes

$$M_i = \frac{\sum_{k \neq i} V_{i,k}}{m_0 - 1}.$$

In the Results section, we compare the $M$-value to the total variance measure on RNA-Seq data from the multi-tissue group experiments, and compare the stably expressed genes identified from these two measures to those identified from microarray data using the CV measure.

## RESULTS

In 'Stably expressed genes', we summarize the stably expressed genes identified from three different experiment groups and emphasize that stability is context dependent. In 'RNA-Seq data collection and processing', we show that traditional house-keeping genes are not necessarily stably expressed according to our numerical measure, and that microarray data and RNA-Seq data may often give different sets of stably expressed genes. In 'Factors affecting stability ranking', we further demonstrate that when using a numerical

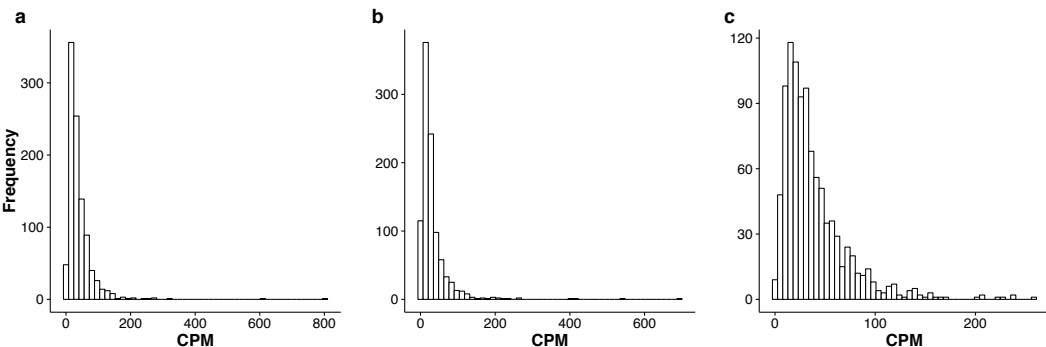

**Figure 2** Histograms of the mean CPM (see Eq. (9)) for the top 1,000 most stably expressed genes identified from the seedling (A), leaf (B) and multi-tissue (C) groups using the total variance measure $\hat{\sigma}^2$. The mean CPM is computed over all samples within each respective group. Note that the $x$ and $y$ axis scales differ between the three plots.

measure to quantify gene expression stability, the outcome will depend on the specific numeric measure used. These points should be intuitive, but they are not often emphasized in practice. In 'Sources of variation', we discuss results from our variance component analysis. In 'Reference gene set for normalization', we discuss how to use the identified stably expressed genes for count normalization.

## Stably Expressed Genes

Using the total variance, $\hat{\sigma}^2$, from the GLMM (see Eq. (6) in 'Poisson log-linear mixed-effects regression model and the total variance measure of expression stability') as a stability measure, we identified stably expressed genes from the three groups of experiments described in 'RNA-Seq data collection and processing': the group of seedling experiments, the group of leaf experiments, and the group of experiments on different tissue types (see Table 1 for a summary). As we mentioned in the Introduction, absolutely stably expressed genes may not exist. Choosing different sample sets as reference allows us to identify stably expressed genes for different biological contexts.

In Tables S2–S4, we summarize the top 1,000 most stably expressed genes in each group. In Fig. 2, we provide the histograms of the mean Count Per Million (CPM) for the 1,000 most stably expressed genes identified in each group. For each gene, the CPM is computed as

$$\frac{\text{count} \times 10^6}{\text{normalized library size}} \tag{9}$$

in each sample and the mean is computed over all samples.

The lists of the top 1,000 genes in the three groups share 104 genes in common (see Table S5 for more details). These genes are stably expressed under a wide range of experimental conditions and in different tissue types, and thus may be worth further study. This list of 104 genes has significant overlap with the top 100 stably expressed genes

identified by *Czechowski et al. (2005)* from a developmental series of microarray samples: 9 out of these 104 genes (see Table S6 for details),

```
AT1G13320, AT1G54080, AT2G20790, AT2G32170,
AT3G10330, AT4G24550, AT5G26760, AT5G46210, AT5G46630,
```

appeared in the list of the top 100 stably expressed genes out of 14,000 genes they examined (the probability is $4.8 \times 10^{-9}$ for a list of 104 genes randomly selected from a set of 14,000 genes to have an overlap of size 9 or more with a pre-selected list of 100 genes). In particular, one gene, AT1G13320, is in all but one of the ten lists of top 500 stably expressed genes identified by *Czechowski et al. (2005)* for different experimental and experimental conditions (the only exception is the set of diurnal series), and is also identified by *Hong et al. (2010)* as a stably expressed gene under all but one of the six experimental conditions they examined. This gene is ranked 159 (top 0.7%), 112 (top 0.5%), 513 (top 2.2%) in the three groups we examined, respectively, according to our stability measure. This gene is a subunit of protein phosphatase type 2A complex and is involved in regulation of phosphorylation and regulation of protein phosphatase type 2A activity. It has been used as a reference gene for normalization in many papers (e.g., *Baron, Schroeder & Stasolla (2012)*; *Bournier et al. (2013)*; these two papers cited *Czechowski et al. (2005)* as reference).

## Comparison to house-keeping genes and stably expressed genes identified from microarray data

*Czechowski et al. (2005)* discussed the expression stability of house-keeping genes and showed that the house-keeping genes are not stably expressed according to their numerical measure. In particular, they compared the expression profiles of five traditional house-keeping genes (AT1G13440, AT3G18780, AT4G05320, AT5G12250, AT5G60390) and five genes (AT1G13320, AT5G59830, AT2G28390, AT4G33380 and AT4G34270) that they identified as stably expressed according to the CV measure from a developmental series of microarray experiments (see Fig.1 of that paper). In Fig. 3, we compare the expression profiles of these 10 genes from *Czechowski et al. (2005)* to the expression profiles of five genes (AT1G64840, AT1G75420, AT2G32910, AT3G51310, AT5G48340) that we randomly selected from the top 100 most stably expressed genes identified from the multi-tissue group RNA-Seq data according the total variance $\hat{\sigma}^2$. For each of the 15 genes, Fig. 3 shows the expression levels measured in CPM over 91 samples in the eight experiments in the multi-tissue group, and Table 2 summarizes the variance components estimated from the GLMM in 'Poisson log-linear mixed-effects regression model and the total variance measure of expression stability'.

The five house-keeping genes show large total variation with all three variance-components relatively large as compared to the other 10 genes. This is consistent with Czechowski's observation that house-keeping genes are not necessarily stably expressed according to a numerical measure. Three of the five stably-expressed genes identified by *Czechowski et al. (2005)* are among the top 1,000 stably-expressed genes according to our stability measure, the total variance $\hat{\sigma}^2$. *Czechowski et al. (2005)* identified those five genes from microarray data and different experiments. It is not too surprising those genes might

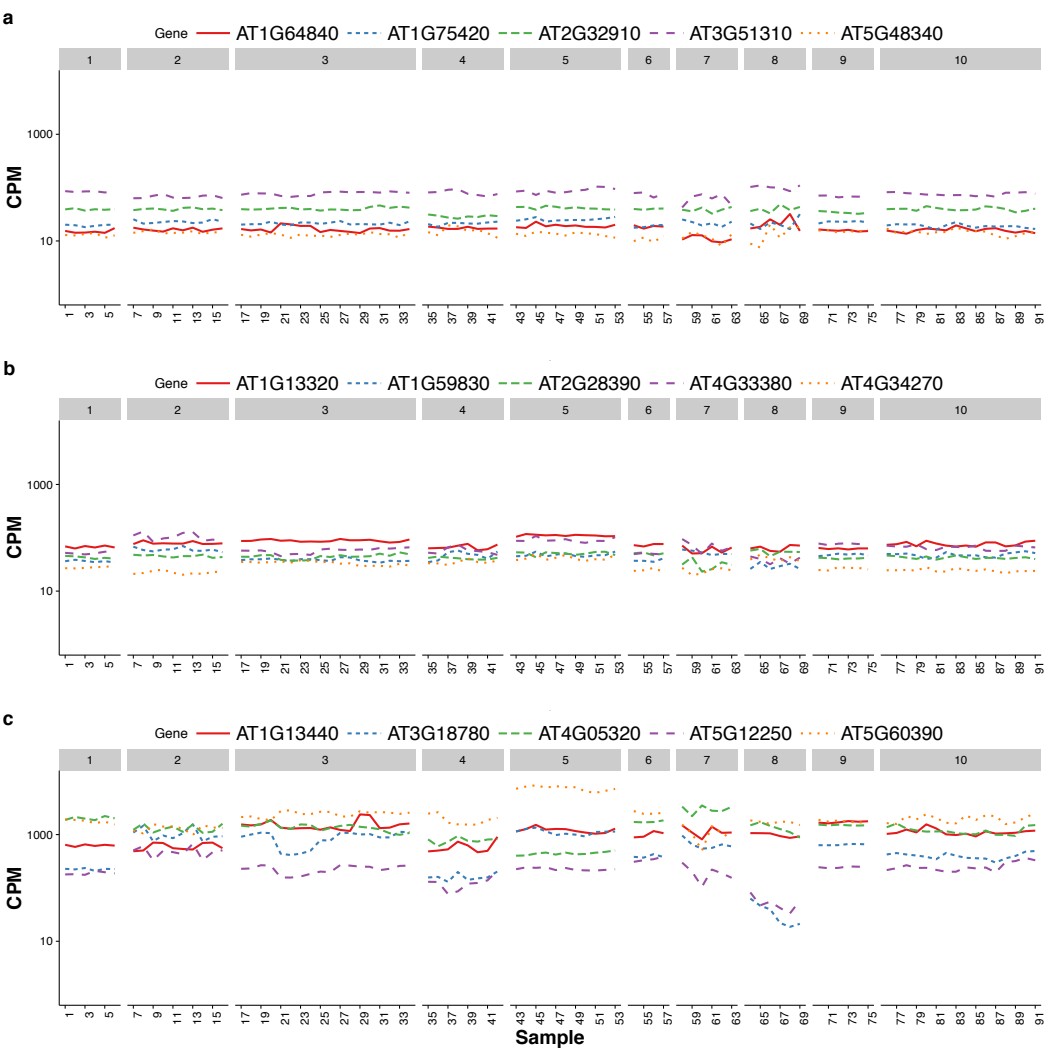

**Figure 3 Expression profiles of 15 genes—as measured by RNA-Seq CPM—across 91 samples in the multi-tissue group.** The 15 genes include (from top to bottom) (A) five stably expressed genes (randomly selected out of the top 100) identified from the multi-tissue group RNA-Seq data using the total variance measure $\hat{\sigma}^2$, (B) five stably expressed identified by *Czechowski et al. (2005)* according to the CV measure from a developmental series of microarray experiments, and (C) five traditional house-keeping genes (HKG) discussed in *Czechowski et al. (2005)*.

not be the most stable in RNA-Seq experiments: the two technologies differ in many aspects including coverage and sensitivity.

The house-keeping genes identified in *Czechowski et al. (2005)* tend to have higher CPM. This is partly due to a selection preference: the authors there intentionally found genes with higher CPM for use as references so that they can be observed in most of the experiments. As we will explain later, we suggest using a collection of 100–1,000 genes as reference gene set for normalization, we did not specifically target for genes with high CPM.

**Table 2 Variance components estimated from the multi-tissue group RNA-Seq data for the 15 genes in Fig. 3 (identified from different sources).** Columns 3–5 are the estimated variance components. Column 6 lists the stability ranking according to the total variance $\hat{\sigma}^2$ in the multi-tissue group.

| Source | Gene | Betweeen-sample | Between-treatment | Between-experiment | Rank |
|---|---|---|---|---|---|
| | AT1G75420 | 0.0012 | 0.0014 | 0.0050 | 5 |
| | AT5G48340 | 0.0042 | 0.0019 | 0.0074 | 46 |
| RNA-Seq | AT2G32910 | 0.0007 | 0.0019 | 0.0113 | 53 |
| | AT1G64840 | 0.0051 | 0.0008 | 0.0095 | 72 |
| | AT3G51310 | 0.0028 | 0.0025 | 0.0100 | 73 |
| | AT2G28390 | 0.0034 | 0.0000 | 0.0111 | 62 |
| | AT1G13320 | 0.0036 | 0.0003 | 0.0258 | 513 |
| Microarray | AT4G34270 | 0.0063 | 0.0000 | 0.0365 | 1074 |
| | AT1G59830 | 0.0044 | 0.0039 | 0.0370 | 1211 |
| | AT4G33380 | 0.0103 | 0.0016 | 0.0747 | 3404 |
| | AT1G13440 | 0.0234 | 0.0058 | 0.1375 | 6562 |
| | AT5G60390 | 0.0267 | 0.0068 | 0.2270 | 8867 |
| HKG | AT4G05320 | 0.0123 | 0.0094 | 0.2690 | 9409 |
| | AT5G12250 | 0.0313 | 0.0128 | 0.3262 | 10589 |
| | AT3G18780 | 0.0375 | 0.0211 | 1.0313 | 14951 |

## Factors affecting stability ranking

The previous two subsections demonstrate that when using a numerical measure to quantify gene expression stability, the outcome is dependent on (1) the biological context reflected in the reference sample set used and (2) the technology used for measuring gene expression. It should also be intuitive, and we will further clarify in the second half of this subsection, that the stability ranking is also dependent on (3) the specific numerical measure used. In this section, we will first compare the lists of stably-expressed genes identified under different scenarios where one or more of the above three factors differ. We then further discuss the subtle roles played by the specific stability measure and the reference gene set by comparing the total variance $\hat{\sigma}^2$ measure from the GLMM (see Eq. (6)) to the $M$-value measure used in the geNorm method (*Vandesompele et al., 2002*).

We look at an additional five lists of stably expressed genes identified under different scenarios and examine how each of these five lists overlaps with the the top stably-expressed genes identified from the multi-tissue group of RNA-Seq experiments according to the total variance measure $\hat{\sigma}^2$ (see 'Poisson log-linear mixed-effects regression model and the total variance measure of expression stability'). The five lists are:

$L_1$: 100 top stably expressed genes from the multi-tissue group according to the $M$-value in geNorm (applied to (count + 1)) of *Vandesompele et al.* ;

$L_2$: 100 top stably expressed genes from the seedling group according to the total variance $\hat{\sigma}^2$ from the GLMM;

$L_3$: 100 top stably expressed genes from the leaf group according to the total variance $\hat{\sigma}^2$ from the GLMM;

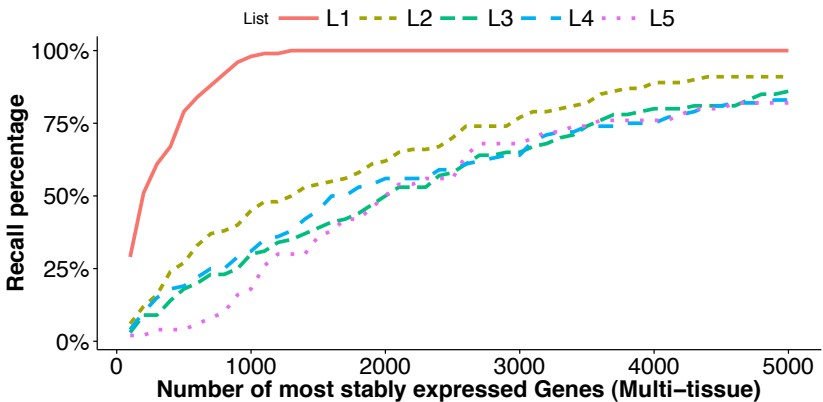

**Figure 4** **Comparison of top stably expressed genes identified under different scenarios.** We choose the top 100 stably expressed genes as described in $L_1$–$L_4$, and the top 50 stably expressed genes in $L_5$ (see 'Factors affecting stability ranking'). and plot the recall percentages between these lists and the top most stably expressed genes identified from the multi-tissue group according to the total variance measure. The $x$-axis is the number of most stably expressed genes in multi-tissue group according to the total variance measure, and the $y$-axis shows the recall percentage (see Eq. (10)) for each of the five lists.

$L_4$: 100 stably expressed genes identified from a developmental series of microarray experiments by *Czechowski et al. (2005)* using the CV measure (see 'Other stability measures');

$L_5$: 50 stably expressed genes identified by *Dekkers et al. (2012)* from microarray seed experiments using the CV measure.

For each list $L_i$ above, we measure how it overlaps with the top stably expressed genes (the reference set) from the multi-tissue group using the *recall percentage*

$$\frac{\#\{L_i \cap \text{reference set}\}}{\#\{L_i\}} \times 100, \tag{10}$$

where #{} denotes the number of elements in the list. In Fig. 4, we plot the recall percentage versus the number of top stably-expressed genes we selected as reference from the multi-tissue group.

We have the following observations:

1. The list $L_1$ is identified from the same set of RNA-Seq experiments as the reference sets, but using a different stability measure ($M$-value in geNorm). This list has significant overlap with the top stably-expressed genes identified using the total variance measure: 29 and 98 out of the 100 genes from the list $L_1$ are among the top 100 and 1,000 most stably-expressed genes, respectively, from the multi-tissue group identified using the total variance measure.

2. The lists $L_2$ and $L_3$ are identified from different sets of RNA-Seq experiments (leaf and seedling experiments) using the same stability measure as used for the reference sets. The lists $L_4$ and $L_5$ are identified from microarray experiments (a developmental series and a seed group) and using the CV measure. The overlapping (recall) percentages are still statistically significant, but much less than in the case of $L_1$. This shows that differences in tissue type and in measuring technology both

influence the expression stability ranking, and to comparable degrees. The lists $L_3$ and $L_5$ have the least overlapping percentages with the reference sets. These lists are identified from a leaf group and a seed group respectively. Our understanding is that the leaf group and the seed group are more biologically homogeneous than the multi-tissue group and thus provide very different biological contexts for evaluating expression stability.

When applied to the same set of samples, the $M$-value and total variance measure $\hat{\sigma}^2$ give similar expression stability ranking: the rank correlation is 0.97 (see also, observation 1 above). We point out that the reason is because the $M$-value and normalization step needed for computing our total variance measure have similar fundamental assumptions. The basic principle behind the $M$-value is that the expression ratio of two stably-expressed genes should be identical in all samples. In formula, it means that the expression values of two stably-expressed genes $i_1$, $i_2$ in any two samples $j_1$, $j_2$ should satisfy

$$\frac{y_{i_1,j_1}}{y_{i_2,j_1}} = \frac{y_{i_1,j_2}}{y_{i_2,j_2}}. \tag{11}$$

Our total variance measure $\hat{\sigma}^2$ is estimated from normalized data. The basic assumption in the normalization step is that majority of genes are not DE. In formula, it means for any stably-expressed gene $i_1$, its expression level as measured by the relative frequency should be stable across all samples,

$$\frac{y_{i_1,j_1}}{S_{j_1}} = \frac{y_{i_1,j_2}}{S_{j_2}}, \tag{12}$$

where $S_{j_1}$ to $S_{j_2}$ are the normalized library sizes (i.e., $R_j N_j$ in Eq. (6)). This implies for any two stably-expressed genes $i_1$ and $i_2$

$$\frac{y_{i_1,j_1}}{y_{i_1,j_2}} = \frac{y_{i_2,j_1}}{y_{i_2,j_2}} = \frac{S_{j_1}}{S_{j_2}}. \tag{13}$$

The first equation in Eq. (13) is equivalent to Eq. (11). (In practical application of both methods, the stability of any single gene is evaluated by comparing its expression to a set of reference genes. See the Method 'Count normalization' for more details.)

In practice, the geNorm program (*Vandesompele et al., 2002*) is frequently used to rank a set of reference genes identified from other methods. An iterative elimination procedure is used along with the $M$-value to determine the final ranks of the expression stability: after each iteration, the gene receiving the largest $M$-value will be removed and a new set of $M$-values will be computed for the remaining genes, and the iteration will go on until there are only two genes left. We did not use such an iterative procedure in the comparisons above (i.e., we only computed one set of $M$-values for all genes). We provided some comments about the iterative elimination procedure in the Appendix.

## Sources of variation

For each gene, the GLMM (Eq. (6) of 'Poisson log-linear mixed-effects regression model and the total variance measure of expression stability') allows us to decompose total count
**Table 3** Percentages—averaged over all genes—of the total variance attributable to each of the three variance components (between-sample, between-treatment, between-experiment) for the three groups of RNA-Seq samples (the seedling, the leaf and the multi-tissue groups).

| Source | Seedling | Leaf | Multi-tissue |
|---|---|---|---|
| Between-sample | 7.2% | 16.0% | 7.6% |
| Between-treatment | 20.1% | 28.0% | 5.1% |
| Between-experiment | 72.6% | 56.0% | 87.3% |

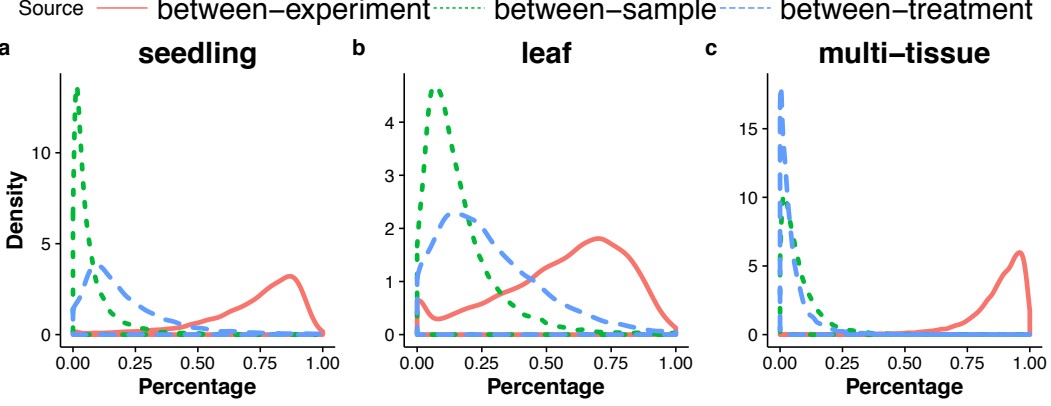

**Figure 5** Distributions (over all genes) of the percentages of the total variance attributable to the between-sample, and between-treatment, or the between-experiment variance component, in the seedling (A), the leaf (B), and the multi-tissue groups (C).

variance into between-sample, between-treatment and between-experiment variance components. The estimated variance components tell us how much each component contributes to the overall count variation. Table 3 summarizes the percentages—averaged over all genes—of the total variance attributable to each of the three components for three groups of RNA-Seq samples (seedling, leaf and multi-tissue groups in 'RNA-Seq data collection and processing'). Fig. 5 shows the histograms of the percentages. Figure 6 shows the stacked bar plot of variance components estimated from the multi-tissue group for 20 genes randomly selected from the top 1,000 stably expressed genes and 20 genes randomly selected from 23,611 genes. As expected, the between-experiment variance component, on average, explains the largest proportion of the total variation. The between-experiment variation is relatively smaller among the leaf samples, indicating that the leaf samples are more homogeneous. There is more variation in the relative percentages of total variance explained by the between-sample and between-treatment variance components. In principle, the between-treatment variation will be greater when there is a higher proportion of DE genes or when the samples are more homogeneous. In practice, the between-sample variance depends greatly on what samples are used as biological replicates.
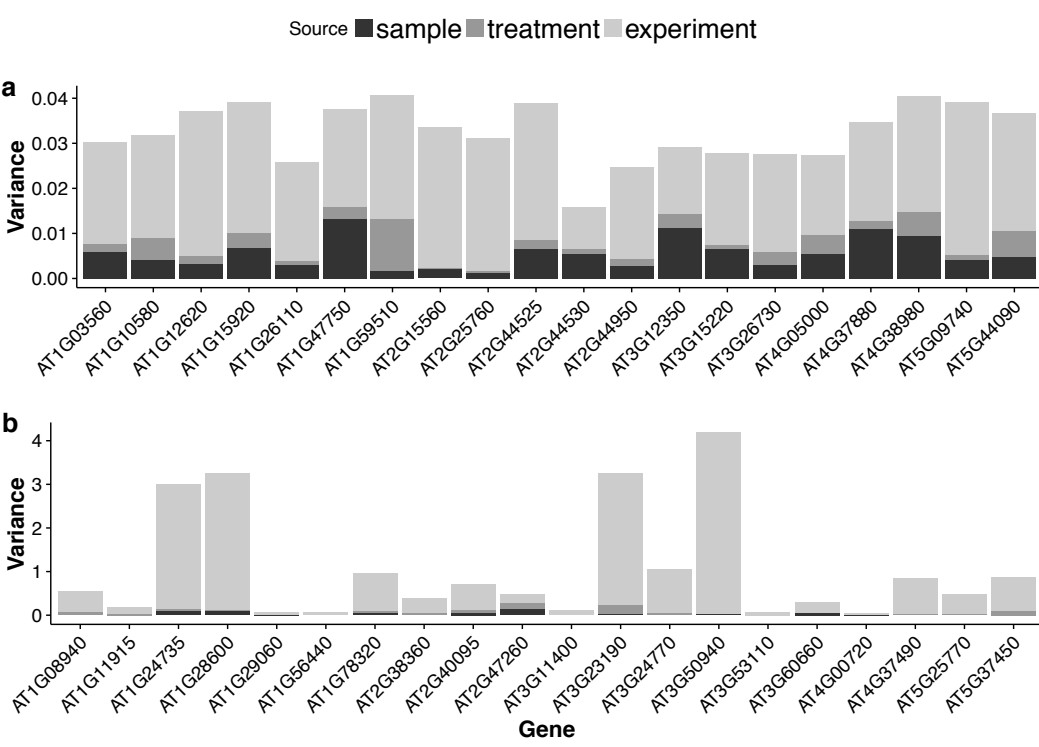

**Figure 6** **Stacked bar plots of the three variance components for selected genes in the multi-tissue group.** (A): 20 genes randomly selected from top 1,000 stably expressed genes; (B): 20 genes randomly selected from all the genes.

## Reference gene set for normalization

Once we have ranked the genes according to our numerical stability measure (i.e., the total variance measure, $\hat{\sigma}^2$), one application is to use an explicit set of most stably expressed genes as reference genes for count normalization. This new approach allows investigators to prescribe a specific biological context for evaluating gene stability by choosing the most relevant reference samples and experiments when computing the stability measure. For example, the most stably expressed genes identified from the multi-tissue group and those identified from the seedling group will provide different biological contexts. In contrast, existing normalization approaches are often applied to the single data set under study, and thus provide a single, narrow context.

Even under a specific biological context, it is almost impossible to know whether the genes in any reference set are absolutely stably expressed, even though commonly used normalization methods often enforce some assumptions on the reference gene set: for example, when we use Anders and Huber's method to estimate the normalization factors based on a subset of reference genes, roughly speaking, the median fold change among the reference genes will be set to 1 (see 'Count normalization' for more details). A subtle point we want to make is that since it is impossible to know how well such or similar assumptions on DE hold for a reference gene set, we can improve the interpretability of the DE test results by making the reference gene set explicit: we can slightly change our perspective and interpret all DE results as relative to the reference gene set. For example, a fold change

Table 4 **A toy example for illustrating the importance of using a common explicit set of reference genes when comparing RNA-Seq data from multiple experiments.** If a common reference gene set (e.g., genes 1–3) is used as reference for count normalization, we will notice that the DE behavior of gene 3 differs in the two experiments. If the two experiments are separately normalized using genes 1–3 as reference in experiment 1, but using genes 3–5 as reference in experiment 2, we may conclude that gene 3 is not DE in either group.

| Gene | Exp. 1 | | Exp. 2 | |
|---|---|---|---|---|
| | **Control** | **Treatment** | **Control** | **Treatment** |
| 1 | 10 | 20 | 10 | 20 |
| 2 | 10 | 20 | 10 | 20 |
| 3 | 10 | 20 | 10 | 10 |
| 4 | 10 | 10 | 10 | 10 |
| 5 | 10 | 10 | 10 | 10 |

of 2 inferred from the GLMM model can be interpreted as the fold change of a gene is 2 times the true (but often unknowable) median fold change of the reference genes. When one estimates the normalization factors based on all genes, one is effectively specifying an implicit set of genes as a reference set. Our proposal is to make the reference set explicit and interpret DE results as relative to the reference gene set.

Interpreting the DE results as relative to an explicit reference set is especially beneficial when one wants to compare DE results from an experiment to previously published results. When the interest is in comparing different experiments, we recommend using a common reference set. For example, when two RNA-Seq data sets are separately normalized with different reference sets, a fold change of two observed in one experiment may not be directly comparable to a fold change of two observed in the other. This concern can be alleviated by using a common set of reference genes. We use a toy example to illustrate this point in Table 4 where we examine the mean counts for 5 genes in two two-group comparison experiments. If we use different reference gene sets for count normalization in the two experiments, for example, we use genes 1–3 as reference in experiment 1, but use genes 3–5 as reference in experiment 2, we may conclude that gene 3 is not DE in either experiment. If we use a common reference gene set—either genes 1–3 or genes 3–5—for normalization, however, we will be able to discover, in either case, that the DE behavior of gene 3 is different in the two experiments. Note that the DE conclusion in both experiments will depend on the reference genes used: if genes 1–3 are used as reference, gene 3 is not DE in experiment 1, but will be DE in experiment 2; if genes 3–5 are used as reference, gene 3 will be considered DE in experiment 1, but not DE in experiment 2. The point is, in either case, we will notice that the DE behavior of gene 3 is different between the two experiments. This information will be lost if one uses different reference sets to assess DE in the two experiments.

In practice, we recommend using the top 1,000 most stably expressed genes for estimating normalization factors. The key is to avoid using too few (e.g., less than 10) or too many (e.g., using all genes) reference genes: intuitively, using too few, the estimates will be unstable; using too many, the results may be subject to influence from highly unstable genes. Our simple simulations suggest that using between 100 to 10,000 genes seems to give stable

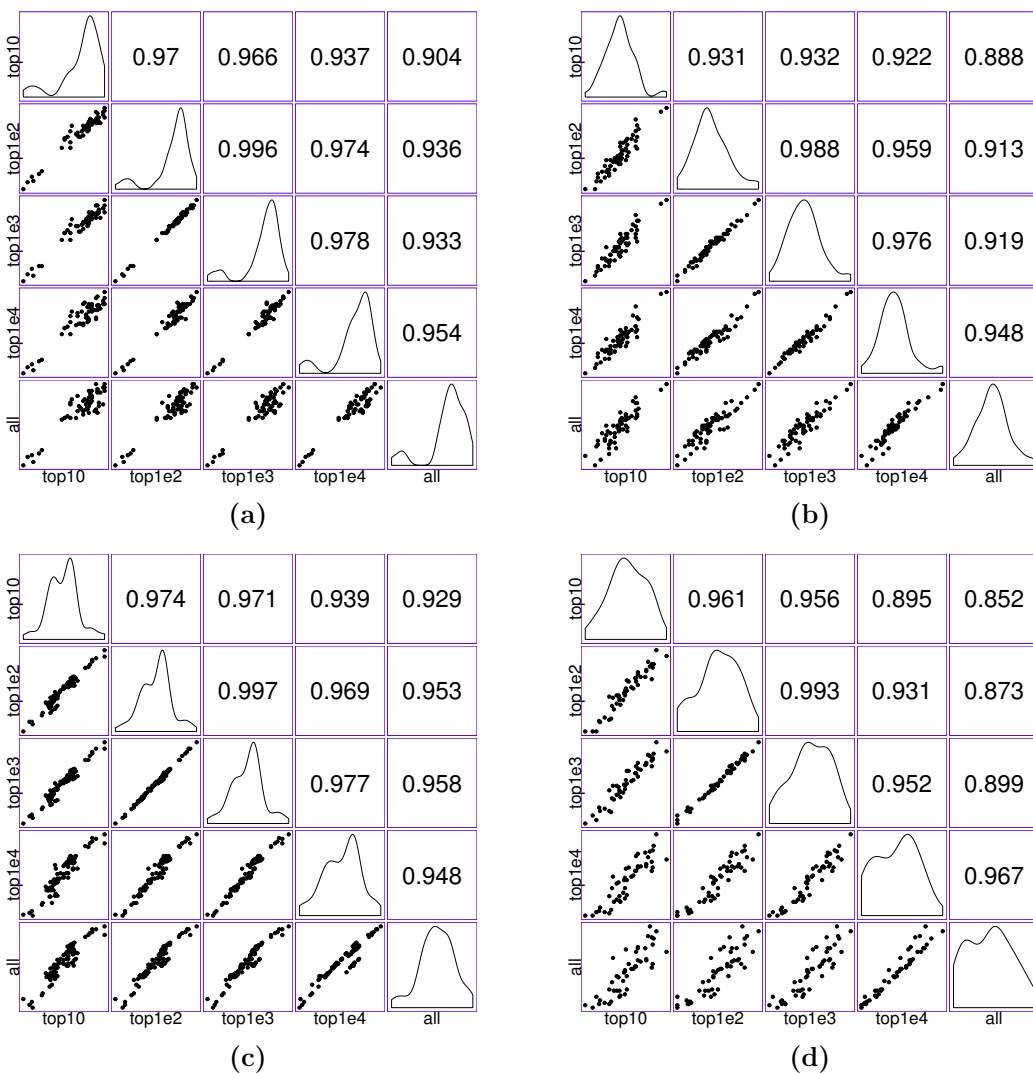

**Figure 7 Matrices of scatter plots of normalization factors estimated using different reference gene sets.** The subfigures (A–C) show normalization factors estimated for the samples in the seedling, leaf, and multi-tissue groups correspondingly. In each case, the top 10, 100, 1,000, and 10,000 stably expressed genes are used as reference to calculate the normalization factors. The subfigure (D) shows the normalization factors estimated for a new root experiment (GSE64410, with sample size 48) using the top 10–10,000 stably expressed genes identified from the multi-tissue group as reference. The normalization factors are estimated using the method described in 'Count normalization'.

results. In the first set of three examples, we used Anders and Huber's method (see Eq. (2)) to estimate normalization factors for samples in each of the seedling, leaf and multi-tissue groups of experiments (see 'RNA-Seq data collection and processing'). We used the top 10, 100, 1,000, and 10,000 stably expressed genes identified earlier (see 'Stably expressed genes' for details) as reference gene sets. Figure 7 shows the pairwise scatter plots and correlation coefficient between the normalization factors when different numbers of top stable genes are used as reference. A stronger correlation indicates the normalization factors estimated from the two settings are highly consistent. The plots and correlation coefficients suggest

using between 100 and 1,000 genes tend to give similar normalization factor estimates. We also used the top 10, 100, 1,000, and 10,000 stably expressed genes identified from the multi-tissue group as reference set for estimating normalization factors for a set of 48 root samples from a new experiment (GSE64410, *Vragović et al. (2015)*). The largest Pearson correlation 0.993 is between the normalization factors estimated using the top 100 and top 1,000 stably expressed genes as reference. Based on the above observations, using 1,000 most stably expressed genes as reference seems to be a reasonable heuristic rule.

### An example

In this part, we illustrate the effect of using different reference gene sets for computing normalization factors on a real data set and explain the implication on DE analysis.

*Wang et al. (2012)* performed RNA-Seq experiments using 10-day old seedlings to investigate the role of Arabidopsis SNW/Ski-interacting (SKIP) protein on transcriptome-wide changes in alternative splicing. Two biological replicates each from wild type (Col-0) and *skip-2* mutant were compared. We retrieved and processed the raw RNA-Seq data from this experiment using our pipeline (see 'RNA-Seq data collection and processing', accession number GSE32216). For this data set, the normalization factors for the four samples (two wild types followed by two mutants) estimated using all genes, $(0.84, 0.62, 1.38, 1.39)$, differ markedly from the normalization factors, $(0.71, 0.54, 1.59, 1.63)$, estimated using the 1,000 reference genes that we identified using the total variance measure from the seedling group (see 'Stably expressed genes').

The implication on DE analysis is that if we use the 1,000 stably expressed genes for normalization, we will expect to see more under-expressed genes and less over-expressed gene in the mutant group relative to the wild type group. The two sets of estimated normalization factors reflect different assumptions: roughly speaking, when using all genes to compute the normalization factors, the assumption is that median fold change among all genes is 1; when using the 1,000 reference genes to compute the normalization factors, the assumption is that the median fold change among the set of 1,000 genes is 1. It is difficult to know which assumption is more reasonable without additional biological insights. However, the benefit of using an explicit set of 1,000 genes as reference is the improved interpretability by making the reference gene set, and thus the implied assumption, more explicit. Furthermore, if one wants to compare the DE results from this experiment to other DE results from the collection of seedling experiments, then one should use a common reference set of genes for count normalization.

## CONCLUSION AND DISCUSSION

In this paper, we advocate quantifying gene expression stability by applying a numerical stability measure to a large number of existing RNA-Seq data sets. Similar strategies have also been used by others to find stably expressed genes from microarray data. Since DE is measured by relative frequencies, we argue that DE is a relative concept and using an explicit reference gene set can improve interpretability of DE results, and furthermore, using a common reference gene set can avoid inconsistent conclusions when comparing multiple experiments (see 'Reference gene set for normalization').

It should be clear but worth emphasizing that when using a numerical measure to identify stably expressed genes, the outcome depends on multiple factors: the background sample set and the reference gene set used for count normalization, the technology used for measuring gene expression, and the specific numerical stability measure used. In this study, to illustrate our proposed methods, we identified three sets of stably expressed genes from three sets of Arabidopsis experiments. The major point is that stably expressed genes identified from different backgrounds will provide different biological contexts for evaluating differential expression. In practice, researchers can choose the specific context. A practical challenge in applying such a philosophy is that no two experiments will have identical settings, and researchers have to decide what experiments can be considered comparable. This is a difficult question; however, we believe it has to be asked from now on: biologists perform comparative experiments with the intent that the conclusions from a single experiment will be generalizable beyond the context of a single lab. If we do not understand comparability between different experiments, such generalization is impossible. Defining and characterizing comparability is a challenging topic that we would like to investigate more in the future.

To identify a set of stably expressed genes, our method still needs to estimate an initial set of normalization factors, which requires that we must make assumptions about relative fold changes between samples. This kind of circular dependence seems unavoidable (*Vandesompele et al., 2002*). In this paper, we used a one-step iteration strategy to reduce the dependence on the initially estimated normalization factors. In future work, we intend to look at the genes through evolutionary genetics methods (e.g., 1,001 genomes, *Weigel & Mott (2009)*). For example, evolutionary genetics methods can help us test whether a gene is under negative, neutral, or positive selection and help us identify genes that are well conserved through the evolutionary history. We need to be mindful that a well conserved gene is not necessarily stably expressed, just like the house-keeping genes. However, it would be interesting to ask whether there is correlation between measures of expression stability and measures of conservativeness, and so on.

In the GLMM model we fit, the random effect terms such as the sample and treatment effects were modeled as normal random variables ('Poisson log-linear mixed-effects regression model and the total variance measure of expression stability'). For the purpose of identifying stably expressed genes, this should be adequate, since we are mainly interested in the variances of these random effects (i.e., the variance components). In the future, it may also be of interest to model these random effects more accurately, for example, in order to build a prior distribution of the random effect terms for analyzing a new data set. A more careful examination of the individual data sets suggests that the between-sample variance varies greatly between experiments. Our observation suggests that different labs often have different understanding of what is deemed as "biological replicates".

The R codes for reproducing results in this paper are available at Github: https://github.com/zhuob/StablyExpressedGenes.

**Table 5  A toy example showing the effect of iterative elimination.** Columns 2 and 3 represent expression levels for seven genes in two samples, column 4 is the stability ranking of genes by $M$-value without iterative elimination, and column 5 is the ranking after two geNorm iterations.

| | Raw counts | | Rank | |
|---|---|---|---|---|
| Gene | Sample 1 | Sample 2 | Rank 1 | Rank 2 |
| Gene1 | 1 | 1 | 3 | 1 |
| Gene2 | 1 | 1 | 3 | 1 |
| Gene3 | 1 | 1 | 3 | 1 |
| Gene4 | 1 | 2 | 1 | 4 |
| Gene5 | 1 | 2 | 1 | 4 |
| Gene6 | 1 | 3 | 6 | |
| Gene7 | 1 | 4 | 7 | |
| Library Size | 7 | 14 | | |

## APPENDIX. THE ITERATIVE ELIMINATION PROCEDURE IN GENORM

In this part, we discuss the effect of an iterative elimination procedure used by geNorm. This iterative elimination procedure creates an extra layer of complexity that is not well explored in literature. We use a toy example below to illustrate one subtle aspect of the iterative elimination procedure. In this example, we consider the expression values of 7 genes in two samples shown in Table 5. When $M$-value is used to rank all 7 genes, the initial ranking of expression stability is given in column 4 of the table: gene 7 is the least stable and genes 4 and 5 are considered the most stable ones. Once genes 6 and 7 are eliminated, however, the recalculated $M$-values will rank genes 1–3 as more stable than genes 4 and 5 (see column 5 of Table 5). The root cause of this reversal of ranking is that when an iterative elimination procedure is used, effectively, the reference gene set is changing after each iteration: in the initial ranking, the expression patterns genes 4 and 5 are close to the "middle of the pack" and thus considered as the most stable, and the expression patterns of genes 1–3 and genes 6 and 7 are considered relatively more extreme; once genes 6 and 7 are removed, however, the "middle of the pack" is shifted towards the expression patterns of genes 1–3, and thus genes 1–3 become the most stably expressed. With this understanding, one could and should make a conscious decision on whether such a behavior as described above is desirable or not.

The essence of the above toy example is that the expression profiles of the set of genes to be ranked are clustered into subgroups. In practice geNorm is often used to rank a set of stably expressed genes. In such applications, the impact of the iterative elimination might be limited. For example, if we use $M$-value to rank the top 1,000 stably expressed genes identified from the multi-tissue group 'Stably expressed genes', the top 100 mostly stably expressed genes from geNorm runs with and without using the iterative elimination will have 77 genes in common.

The point we want to emphasize is that gene stability is a relative concept and the stability ranking depends on which set of genes we use as reference. In an iterative elimination

procedure, the reference gene set will change after each iteration. The procedure can thus give surprising results and the adoption of it in practice should not be automatic.

## ACKNOWLEDGEMENTS

We thank Duo Jiang and Wanli Zhang for helpful discussions. This article is part of a doctoral dissertation written by BZ under the supervision of YD.

### Funding
Research reported in this publication was supported by the National Institute of General Medical Sciences of the National Institutes of Health under Award Number R01GM104977 (to YD, SCE, and JHC). The funders had no role in study design, data collection and analysis, decision to publish, or preparation of the manuscript.

### Grant Disclosures
The following grant information was disclosed by the authors:
National Institute of General Medical Sciences of the National Institutes of Health: R01GM104977.

### Competing Interests
Jeff H. Chang is an Academic Editor for PeerJ.

### Author Contributions
- Bin Zhuo performed the experiments, analyzed the data, wrote the paper, prepared figures and/or tables, reviewed drafts of the paper.
- Sarah Emerson wrote the paper, reviewed drafts of the paper.
- Jeff H. Chang conceived and designed the experiments, reviewed drafts of the paper.
- Yanming Di conceived and designed the experiments, contributed reagents/materials/-analysis tools, wrote the paper, reviewed drafts of the paper.

### Data Availability
The raw data sets at GitHub are seedling.rds, leaf.rds , tissue.rds; https://github.com/zhuob/StablyExpressedGenes/tree/master/R.

### Supplemental Information
Supplemental information for this article can be found online at http://dx.doi.org/10.7717/peerj.2791#supplemental-information.

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
