# Peer review of "Identifying stably expressed genes from multiple RNA-Seq data sets"

_PeerJ, doi:10.7717/peerj.2791_

## Round 0.1 · original submission · Minor Revisions

Address all the queries referees have raised for the improvement of the manuscript.

A question from myself: Are there any differences because of tissue type selections?

·

Basic reporting

No Comments

Experimental design

No Comments

Validity of the findings

No Comments

Additional comments

Authors suggest selecting explicit reference sets and choosing between 100 to 1000 stably expressed genes as reference set. The authors base these recipe on the observations they made. But as seen by the recall percentages (Page 10), the reliability is low in certain cases (L3 (different biological context) and L5 (technology platform)). Can this be improved by the iterations to calculate the normalization factors?

Although a negative effect of geNorm iterative procedure is that it changes the rank order of the stably expressed genes, they all might be still within the top N in the list.
It may be useful to test how the recall percentages compare, using the 5 lists using the iterative process for the numerical measure used by the authors.

Reviewer 2 ·

Basic reporting

1. After filtering those genes with low read counts, the last column in Table 1 is the number of remaining genes. How many genes overlap between the two groups ? How many genes overlap among the three groups ?

2 Page8Line292. These five genes (AT1G63110, AT1G79280,293 AT3G27530, AT4G02560, AT5G53540) are not in the Table 2 and Figure 2.

Experimental design

1. Page5 Line166. The multi-mapping or multi-overlapping reads were not counted. However, the multi-mapping and multi-overlapping reads also contain some useful information in RNA-seq data. Why do you discard these reads in your study? Many existing applications can handle these reads, such as HTSeq.

2. The stably expressed genes are useful for count normalization and differential expression analysis. In the whole paper, you focus on the count normalization. After count normalization, you may further validate your thought by DE analysis using real-world RNA-seq data.

Validity of the findings

1. In Table 2, the five house-keeping genes show large total variation compared to other 10 genes. And in figure 2, the five house-keeping genes usually have high CPM values than other 10 genes. Why these stably expressed genes from RNA-seq data are relatively small CPM values? In the top 100 stably expressed genes, whether there are genes contain CPM values beyond 1000? How many?

Reviewer 3 ·

Basic reporting

In this paper authors examines and reports factors that effect the identification of stably expressed genes in RNA-seq data, and introduce another stability measure. Authors emphasize that gene stability is a relative concept and stability ranking depends of the reference gene set. They suggest identification of a explicit reference gene set for normalization when combining multiple RNA-seq experiments. On the other hand, they clearly show that differential expression analysis results might change dramatically depending on selected reference set.

In general, the paper is well written. It is easy to read and understand.

Minor:
Abstract should note that all RNA-seq is from Arabidopsis experiments and they used three different tissue types (sample categories etc.)

Experimental design

Methods are described in detail.

Validity of the findings

Minor:
Lines 358-371: Authors give a toy example to explain a potential drawback of geNorm algorithm. This algorithm, as well as proposed algorithm, assumes evaluation of large number of gene features. Plotted scenario would never happen when 1000s of gene features used. I suggest removal of that discussion.

---

## Round 0.2 · accepted · Accept

Congratulations!!!

Please work with our production team if any changes to be included in the final version.

Reviewer 2 ·

Basic reporting

The authors had been revised the paper according to reviewers, and the paper is well written.

Experimental design

No Comments

Validity of the findings

No Comments

Additional comments

No Comments